# The Spatiotemporal Evolution and Prediction of Carbon Storage: A Case Study of Urban Agglomeration in China's Beijing-Tianjin-Hebei Region

Yingting He [1], Chuyu Xia [2], Zhuang Shao [1] and Jing Zhao [1,*]

[1] School of Landscape Architecture, Beijing Forestry University, Beijing 100083, China; yingting_he@bjfu.edu.cn (Y.H.); no81terry@bjfu.edu.cn (Z.S.)

[2] Faculty of Architecture, Civil and Transportation Engineering, Beijing University of Technology, Beijing 100124, China; chuyu.xia@bjut.edu.cn

[*] Correspondence: zhaojing@bjfu.edu.cn

**Abstract:** Due to rapid urban expansion, urban agglomerations face enormous challenges on their way to carbon neutrality. Regarding China's urban agglomerations, 25% of the land contains 75% of the population, and all types of land are used efficiently and intensively. However, few studies have explored the spatiotemporal link between changes in land use and land cover (LULC) and carbon storage. In this work, the carbon storage changes from 1990 to 2020 were estimated using the InVEST model in China's Beijing–Tianjin–Hebei (BTH) region. By coupling the Future Land Use Simulation (FLUS) model and InVEST model, the LULC and carbon storage changes in the BTH region in 2035 and 2050 under the natural evolution scenario (NES), economic priority scenario (EPS), ecological conservation scenario (ECS), and coordinated development scenario (CDS). Finally, the spatial autocorrelation analysis of regional carbon storage was developed for future zoning management. The results revealed the following: (1) the carbon storage in the BTH region exhibited a cumulative loss of $3.5 \times 10^7$ Mg from 1990 to 2020, and the carbon loss was serious between 2000 and 2010 due to rapid urbanization. (2) Excluding the ECS, the other three scenarios showed continued expansion of construction land. Under the EPS, the carbon storage was found to have the lowest value, which decreased to $16.05 \times 10^8$ Mg in 2035 and only $15.38 \times 10^8$ Mg in 2050; under the ECS, the carbon storage was predicted to reach the highest value, $18.22 \times 10^8$ Mg and $19.00 \times 10^8$ Mg, respectively; the CDS exhibited a similar trend as the NES, but the carbon storage was found to increase. (3) The carbon storage under the four scenarios was found to have a certain degree of similarity in terms of its spatial distribution; the high-value areas were found to be clustered in the northwestern part of Beijing and the northern and western parts of Hebei. As for the number of areas with high carbon storage, the ECS was found to be the most abundant, followed by the CDS, and the EPS was found to be the least. The findings of this study can help the BTH region implement the "dual carbon" target and provide a leading example for other urban agglomerations.

**Keywords:** land use; carbon storage; FLUS-InVEST model; spatial autocorrelation analysis; multi-scenario simulation

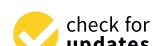



## 1. Introduction

Terrestrial ecosystems, as the key carbon reservoirs of the carbon cycle, are intimately linked to human production and life. Land use and land cover (LULC) changes are responsible for one-third of global anthropogenic carbon emissions, which in turn affect changes in carbon sources and sinks [1,2]. Therefore, the carbon cycle-related responses of terrestrial ecosystems and global climate change driven by carbon emissions have come to the forefront of contemporary research. Optimizing land-use patterns through rational adjustment of land-use types is the key to achieving carbon emission reduction [3,4].

In China, urban agglomerations account for 75% of the population, while occupying 25% of the land, and all types of land are used efficiently and intensively. Studies have shown that the decrease in vegetation cover and the increase in urban construction land have given rise to a severe loss in urban carbon storage [5]. In particular, in areas characterized by the rapid expansion of urban construction land, such as urban agglomerations and mega-cities, the loss of carbon storage is serious [6,7]. The "dual carbon" target of carbon peaking and carbon neutrality goals is a pivotal strategy for China today [8]. With the accelerated urbanization process, the change in the carbon sink capacity caused by LULC directly affects the achievement of the "dual carbon" target. According to relevant national policies, the Beijing–Tianjin–Hebei (BTH) region will carry out the construction of carbon-neutral demonstration zones and take the lead in exploring carbon-neutral development paths [9]. The "dual carbon" route was identified as the main task in the Report on the Work of the Beijing Government in 2021, the Regulations on the Promotion of Carbon Peaking and Carbon Neutrality in Tianjin, and the Implementation Opinions of Hebei. In addition, documents such as the Report on the Work of the State Council in 2021, the Beijing Urban Master Plan (2016–2035), the Tianjin Territorial Spatial Master Plan (2021–2035), and the Hebei Territorial Spatial Plan (2021–2035) all propose the improvement of the carbon sink capacity of urban systems via various means, such as territorial spatial planning, to aid in the implementation of the "dual carbon" target.

The BTH region is characterized by a high population density and rapid urbanization, which have brought about land-use changes such as the significant increase in construction land [10]. The total carbon emissions of the BTH region account for up to one-fifth of the country's total, making it the world's largest urban agglomeration in terms of carbon emissions [11]. The realization of the integrated and coordinated development of the BTH urban agglomeration's low-carbon economic transformation is China's current focus, and the implementation of the "dual carbon" target of the BTH region is necessary to achieve the target on schedule.

It is necessary to evaluate the past and predict future land-use changes in order to optimize low-carbon development. The carbon storage of the BTH region can be calculated through reliable data and scientific methods, and the relationship between the changes in LULC and carbon storage can be discussed. In future land-use simulations under different scenarios, many models have been extensively used, such as the CA-Markov model [12], the CLUE-S spatiotemporal model [13], the Future Land Use Simulation (FLUS) model [14], and the Patch-generating Land Use Simulation (PLUS) model [15]. Houghton's bookkeeping model has been the most influential in the study of ecosystem carbon cycles [16]. Among the many ecosystem service models, the Integrated Valuation of Ecosystem Services and Trade-offs (InVEST) model is the most widely applied due to its simple quantification and strong visibility [17]. It has been successfully applied in China [18], California [19], Sri Lanka [20], Morocco [21], Sariska [22], Pakistan [3], and other countries and regions. There currently exists a foundation for the study of linking future land use and carbon storage changes. For example, at the urban scale, based on climate change scenarios, Wang et al. simulated the land-use changes and estimated the carbon storage in Bortala, China [23]. At the provincial scale, Liu et al. assessed and predicted the response of carbon storage to LULC on Hainan Island [24]. At the urban agglomeration scale, Jiang et al. simulated the possible consequences of ecosystem changes on carbon storage in three urban sprawl scenarios in the Changsha–Zhuzhou–Xiangtan urban agglomeration [18]. At the watershed scale, Zhao et al. assessed the ecological engineering's effects on carbon storage in the Heihe River Basin's upper reaches [25]. At the scale of a specific region, Rafael et al. projected changes in land use to carbon storage in a Mexican western basin [26]. Research on urban zoning management models based on spatial autocorrelation techniques also has a certain basis, such as Xia et al. analyzed the local spatial autocorrelation of carbon emissions from daily travel to study the local differences in the impact of urban form on carbon emissions [27]. Regarding the BTH region, some scholars have studied its past land-use changes or its relationship with carbon sinks [28–30]. Other scholars have run simulations

of the future urban landscape dynamics under different scenarios [31,32] or have conducted multi-scenario simulations of the land use in specific regions [33]. Above all, we could conclude from the present study that land-use type change is the main driving factor of carbon storage, but current studies mainly simulated regional carbon storage by focusing on a single method or a single scenario, while few studies have linked land use and carbon storage and conducted multi-scenario simulations of the future urban pattern to investigate the relationship. It is of vital importance to give priority to relieving Beijing of non-essential functions due to its status as the capital when pursuing the integrated development of the BTH region. By considering this as a breakthrough point and subsequently implementing future zoning management, focus can be placed on the long-term development of the urban agglomeration.

To sum up, taking the BTH region as the study area, this work looked at land-use changes and their effects on the carbon storage spatiotemporally in different periods from 1990 to 2020. A multi-scenario simulation of the urban agglomeration patterns of the BTH region in 2035 and 2050 was also conducted. The findings provide the BTH region a scientific basis for the territorial spatial planning under the "dual carbon" target, and for the promotion of the BTH's development from intermediate to advanced coordination. Moreover, based on the findings, the BTH region can provide a leading and demonstrating role for other urban agglomerations in China.

## 2. Materials and Methods

### 2.1. Study Area

The Beijing–Tianjin–Hebei (BTH) urban agglomeration (113°04′ E–119°53′ E, 36°01′ N–42°37′ N) covers a total area of 218,000 km$^2$. It encompasses the municipalities of Beijing and Tianjin, as well as the province of Hebei (Chengde, Zhangjiakou, Baoding, Shijiazhuang, Qinhuangdao, Hengshui, Cangzhou, Xingtai, Langfang, Tangshan, and Handan), with 13 cities under its jurisdiction. The overall topography of the BTH region slopes in a step-like manner from northwest to southeast, and is situated at the northernmost of the North China Plain. There are various types of landforms in the region, including plains, mountains, and hills. The geographic location and elevation map are shown in Figure 1. Located on the coast of the Bohai Sea, the BTH has temperate monsoon climate with rainfall concentrated in the summer, an annual average temperature of 0–12 °C, and annual average precipitation of 400–800 mm. The vegetation is mainly temperate broad-leaved forests. With the promotion of projects such as the Three-North Shelter Forest Programme and the Beijing–Tianjin Sandstorm Source Control Project [34], as of 2020, the forest coverage rates of Beijing, Tianjin, and Hebei, respectively, reached 43.77, 12.07, and 26.78%. The BTH is China's third-largest urban agglomeration, serving as the country's political, cultural, and international exchange hub, as well as a hotbed of scientific and technological innovation. Besides, it is also "a world-class urban agglomeration with the capital as the core, a new national innovation-driven economic growth engine, a pioneer area of regional coordinated development and reform, and a demonstration area for ecological restoration and environmental improvement". Nevertheless, with the rapid acceleration of urbanization, increasingly more ecological land is being converted into construction land, thus posing a challenge to the BTH's sustainable development [35].

### 2.2. Data Sources

#### 2.2.1. Land-Use Data

Data on the LULC and the BTH administrative boundary with a spatial resolution of 100 m × 100 m were downloaded from the Resource and Environment Science and Data Center. Seven periods of LULC data from 1990 to 2020 (1990, 1995, 2000, 2005, 2010, 2015, and 2020) were selected. The Kappa test was used to verify the classification quality and the overall accuracy of the dataset was found to be above 90%. ArcGis10.6 software was used to divide the data on the land-use types in this region into 25 secondary types. Six

first-level land types were formed via reclassification, including cultivated land, woodland, grassland, water area, construction land, and unused land.

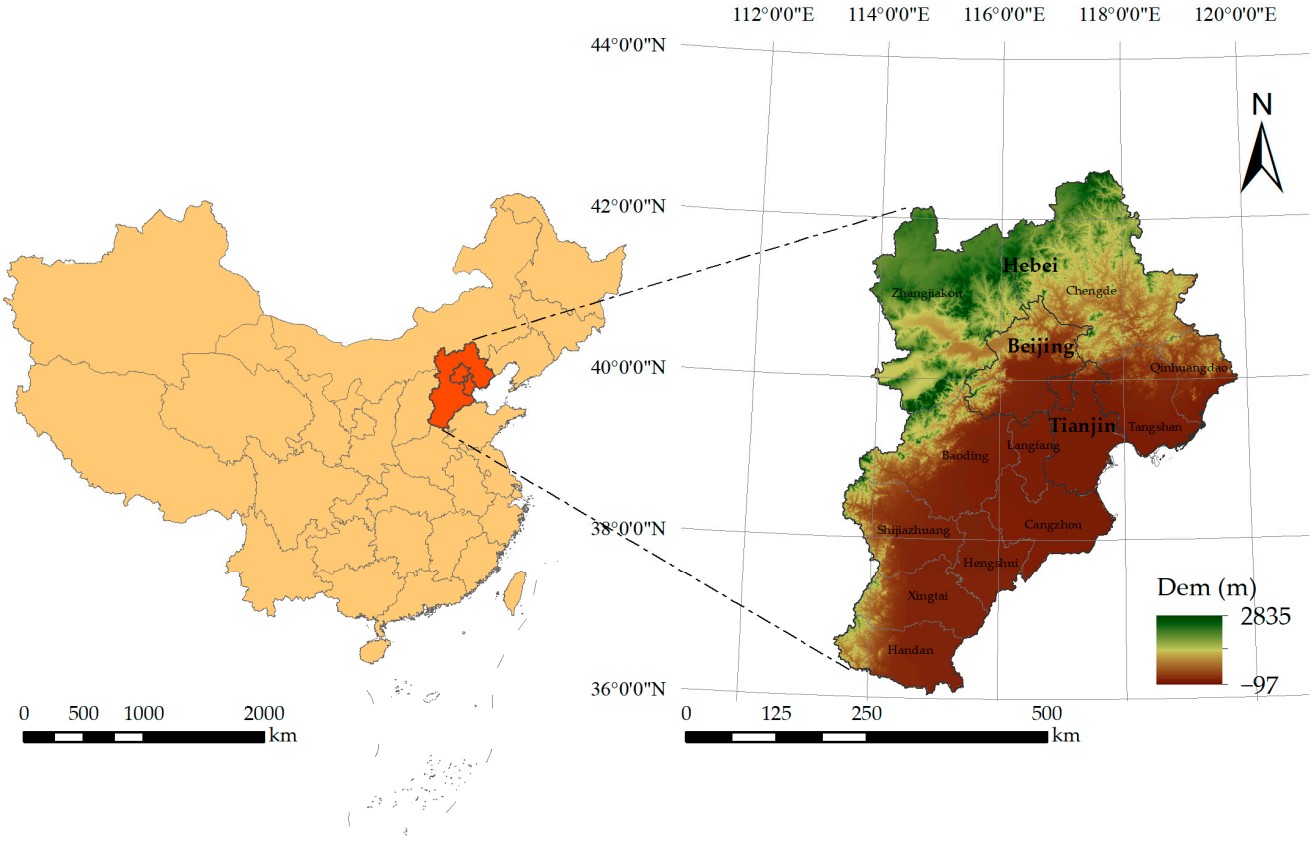

**Figure 1.** The geographical location and elevation map of the BTH region.

2.2.2. Driving Factors Data

The natural conditions, socioeconomic factors, and planning policies were classified as driving factors for land-use forecasting: natural factors, included the DEM (Digital Elevation Model), MAP (Mean Annual Precipitation), AMT (Annual Mean Temperature), and NDVI (Normalized Difference Vegetation Index). Socioeconomic factors included the distance to town, transportation network, GDP, and population density. Planning factors included the urban development boundary, permanent basic agricultural land, and ecological protection red line [36]. The data are shown in Table 1. Using the resampling function of AcrGIS10.6, the resolution of land-use driving factors was uniformly adjusted to 100 m × 100 m.

**Table 1.** The land-use drivers and data sources.

| Data Type | Data Name | Resolution (m) | Data Source |
|---|---|---|---|
| Natural factors | NDVI | 500 | MOD13Al Version 6 (http://search.earthdata.nasa.gov/, accessed on 18 January 2022) |
| | DEM | 90 | Resource and Environment Science and Data Center (http://www.resdc.cn, accessed on 28 February 2022) |
| | MAP | 1000 | |
| | AMT | 1000 | |

**Table 1.** *Cont.*

| Data Type | Data Name | Resolution (m) | Data Source |
|---|---|---|---|
| Socioeconomic factors | Distance to town | 1000 | National Geomatics Center of China (http://www.ngcc.cn/ngcc/, accessed on 26 February 2022) |
| | Transportation network | 30 | OpenStreetMap (http://www.openstreetmap.org/, accessed on 26 February 2022) |
| | GDP<br>POP | 1000<br>1000 | Resource and Environment Science and Data Center (http://www.resdc.cn, accessed on 28 February 2022) |
| Planning factors | Ecological red line | 100 | Beijing Municipal Commission of Planning and Natural Resources (http://ghzrzyw.beijing.gov.cn/, accessed on 29 January 2022) |
| | Permanent basic farmland | 100 | Tianjin Municipal Bureau of Planning and Natural Resources (http://ghhzrzy.tj.gov.cn/, accessed on 29 January 2022) |
| | Urban growth boundary | 100 | Department of Natural Resources of Hebei Province (http://zrzy.hebei.gov.cn/, accessed on 29 January 2022) |

NDVI: Normalized Difference Vegetation Index; DEM: Digital Elevation Model; MAP: mean annual precipitation; AMT: annual mean temperature; GDP: gross domestic product; POP: population.

*2.3. Methods*

This research consisted of five main steps. The first step was to estimate the carbon storage changes in each period based on the LULC data for seven periods from 1990 to 2020 using the InVEST model. Secondly, four future LULC scenarios were set up according to the past change patterns, with planning policies as a platform. Under four scenarios, the FLUS model was assembled in the third step to project the future land-use changes. The spatiotemporal distributions of carbon storage in 2035 and 2050 under these four scenarios were assessed and compared with the results of the InVEST model in the fourth step. Finally, a spatial autocorrelation analysis of regional carbon storage was performed with the assistance of GeoDa. The research framework is shown in Figure 2 below.

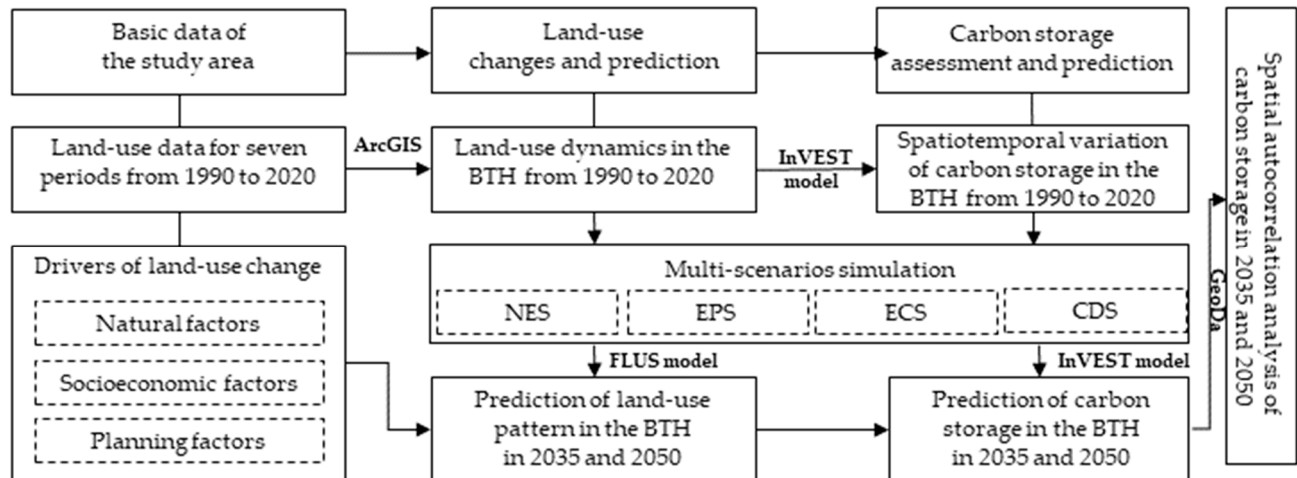

**Figure 2.** Research framework.

2.3.1. Carbon Storage Assessment with the InVEST Model

The InVEST model aims to weigh the link between land use and ecosystem service functions. The Carbon Storage and Sequestration module aggregates the biophysical amount of carbon stored in four carbon pools, namely the aboveground living biomass (i.e., carbon in all living plant material above the soil), belowground living biomass (i.e., the

carbon present in living root systems of aboveground biomass), soil organic matter (i.e., carbon distributed in the organic component of soil), and dead organic matter (i.e., carbon in the litter as well as lying and standing deadwood). Rooted on different land-use types and carbon densities, the carbon density for LULC type $i$ can be expressed as follows:

$$C_i = C_{ia} + C_{ib} + C_{is} + C_{id}, \tag{1}$$

$$C_{tot} = \sum_{i=1}^{n} C_i \times S_i, \tag{2}$$

where $i$ represents the land-use type, $C_i$ represents the total carbon density of land-use type $i$, $C_{ia}$ represents the aboveground density, $C_{ib}$ represents the belowground density, $C_{is}$ represents the soil organic carbon density, and $C_{id}$ represents the dead organic carbon density. The unit of all carbon densities is megagrams per hectare (Mg/ha). Additionally, $C_{tot}$ represents the total carbon storage of the ecosystem (Mg), $S_i$ represents the area of land-use type $i$ (ha), and $n$ represents the number of land-use types; in this study, $n = 6$ [37].

According to the current situation of the BTH region, by comparing it with a region at the same latitude and drawing on previous research results and experience [38–40], the carbon density coefficients of the four basic carbon pools in the research area are listed in Table 2.

**Table 2.** The carbon storage per unit area of each LULC type.

| Land-Use Type | $C_{above}$ | $C_{below}$ | $C_{soil}$ | $C_{dead}$ |
|---|---|---|---|---|
| Cultivated land | 3.99 | 0.00 | 32.90 | 0.00 |
| Woodland | 41.69 | 20.99 | 126.32 | 1.95 |
| Grassland | 20.49 | 16.47 | 51.33 | 22.06 |
| Construction land | 3.64 | 1.82 | 6.25 | 0.57 |
| Water area | 2.06 | 0.52 | 78.64 | 0.10 |
| Unused land | 0.00 | 0.00 | 0.00 | 0.00 |

2.3.2. Future Land Use Simulation (FLUS) Model

The FLUS model is composed of an artificial neural network (ANN) and an adaptive inertia competition mechanism. The ANN is capable of revealing the relevance between the social, economic, and natural elements and land-use changes. The complexity and uncertainty of mutual transformation can be overcome by the adaptive inertia competition mechanism, hence remedying the parameter determination in traditional cellular automata and the complexity of local conversion.

The accuracy was first verified. According to the land-use change pattern of the BTH region from 1990 to 2020, the land-use situation in 2020 was obtained after simulation and compared with the actual situation. The accuracy of land pattern evolution prediction was tested using the Kappa coefficient [41]—the land-use data in 2015 were used as the training set to project the land-use pattern in 2020 for the validation of the model, with a Kappa coefficient of 90.10% and an overall accuracy of 91.27%. This shows that the FLUS model had good simulation capability and high accuracy at the scale of the BTH region; when Kappa > 0.8, the result is credible, the simulation effect is excellent, and the degree of consistency is excellent as well; when 0.6 < Kappa ≤ 0.8, the simulation effect and the degree of consistency are both good; when 0.4 < Kappa ≤ 0.6, the simulation effect is effective and the degree of consistency is moderate; when 0.2 < Kappa ≤ 0.4, the simulation effect is poor; when Kappa ≤ 0.2, the simulation effect is extremely poor [42].

Based on the BTH region's land-use type maps for seven periods from 1990 to 2020 and the selected drivers affecting land-use changes, the land-use types were fitted with the drivers in the suitability probability estimation module of the ANN. Each land-use type's suitability probabilities in 2035 and 2050 under four scenarios were calculated.

In view of other related papers, the parameters were fine-tuned and several experiments were conducted to set the neighborhood weights of each factor under the four scenarios, as shown in Table 3 [43,44].

| Land-Use Type | Cultivated Land | Woodland | Grassland | Water Area | Construction Land | Unused Land |
|---|---|---|---|---|---|---|
| Neighborhood weight | 0.6 | 1 | 0.3 | 0.1 | 0.2 | 0.1 |

The adaptive inertia competition mechanism of the FLUS model, which is grounded on roulette selection, was employed to address the uncertainty and complexity of multiple land-use types that are interconverted owing to the combined influence of human activities and natural effects based on which the comprehensive rules were calculated.

$$I_k^t = \begin{cases} I_k^{t-1}, \left(if \ |D^{t-2}| \leq |D^{t-1}|\right) \\ I_k^{t-1} \times \frac{D^{t-2}}{D^{t-1}}, \left(if \ 0 > D^{t-2} > D^{t-1}\right) \\ I_k^{t-1} \times \frac{D^{t-1}}{D^{t-2}}, \left(if \ D^t - 1 > D^{t-2} > 0\right) \end{cases}, \tag{3}$$

where $I_k^t$ is the inertia coefficient when the number of iterations is $t$, and $D_k^{t-1}$ is the difference in pixels between the predicted and the actual land-use type when reaching $t - 1$.

### 2.3.3. The Setting of Different Future Scenarios

Multiple factors influence the future development and land-use changes of cities, therefore, a variety of environmental influences must be fully considered when simulating and predicting the land-use changes [45]. On account of the substantial differences between the urban development levels Beijing, Tianjin, and Hebei, the Beijing Urban Master Plan (2016–2035), the Tianjin Territorial Spatial Master Plan (2021–2035), and the Hebei Territorial Spatial Plan (2021–2035) were considered as the overall guiding documents in this study. Additionally, by considering the BTH urban agglomeration as the spatial scale and the current situation of urbanization, the following four scenarios are proposed: the natural evolution scenario (NES), economic priority scenario (EPS), ecological conservation scenario (ECS), and coordinated development scenario (CDS). The land-use distributions of the BTH region in 2035 and 2050 were respectively simulated and predicted, and were finally combined via ArcGIS.

1. Natural Evolution Scenario (NES)

The binding influence of any planning policy on land-use changes is not considered, and the rate of change from 1990 to 2020 is used as the reference basis for the land simulation of the future scenario under natural conditions.

2. Economic Priority Scenario (EPS)

The population increases rapidly and the GDP grows rapidly. The area of ecological land such as grassland, cultivated land, as well as woodland shrinks, while the area of construction land expands rapidly. By 2035, the population distributions of Beijing, Tianjin, and Hebei will increase by 8, 40, and 12%, respectively; the GDP will grow by 80, 95, and 83%. By 2050, the capacity of urban expansion will be decreased, as will the growth rate of all indicators.

3. Ecological Conservation Scenario (ECS)

According to the requirement of protecting the substrate, ecological protection takes priority over economic development. Planning is considered the overall guiding document and unified action program for the construction of urban clusters. In accordance with the requirements of ecological function zoning and main function zoning, the city cluster is considered the spatial scale, the integration of regional ecological construction and the homogeneous construction of landscape ecological structure is promoted, and the goal of the ecological co-construction and sharing of a shared natural environment is realized.

Beijing: Based on the red lines of ecological and permanent basic farmland protection, the forest coverage rate will account for 45% of the city area by 2035. Additionally, the proportion of the city's ecological control area will increase to 75% by 2035, and to more than 80% by 2050. The probability of converting various types of land into construction land will be reduced, the conversion of water bodies into construction land will be prohibited, and an ecological system with visible mountains and good quality will be constructed.

Tianjin: The permanent basic farmland and ecological red line will be used as restricted conversion zones, and the structures of multiple land-use types will be comprehensively considered. The amount of construction land will be controlled, and the conversion probabilities of four important ecological land-use types will also be adjusted. The rational use of valuable land resources will be integrated into the Beijing-Tianjin-Hebei Capital Ring Ecological Barrier Belt to optimize the regional ecological pattern.

Hebei: The permanent basic farmland and ecological red line will be used as restricted conversion areas. In accordance with the guarantee of a moderate and reasonable scale and stability, the probability of the conversion of each type of land to construction land will be reduced to implement its positioning as the "Beijing–Tianjin–Hebei ecological environment support area".

4. Coordinated Development Scenario (CDS)

Under the premise of strictly protecting high-quality cultivated land and the ecological environment in the BTH, spatial pattern optimization will lead to the coordinated development of the land-use tasks. The urban growth boundary will be the restricted area, and the three major metropolitan areas, namely the "Capital Metropolitan Area", "Tianjin Metropolitan Area", and "Shijiazhuang Metropolitan Area", will be the tripod that supports the high-quality development of the BTH. Thus, the blossom of the whole BTH city cluster will be driven by radiation, and will be promoted from intermediate to advanced coordination.

Beijing: The reduction of urban and rural construction land will be promoted to improve its quality and intensive and efficient use, rooted in the development needs of Beijing. The scale of urban construction land will be limited to 2760 km$^2$ by 2035, and the probability of each category being converted to this land-use type will be reduced. Non-capital functions will be decentralized and closely dovetailed with the BTH coordinated development strategy.

Tianjin: Grounded on the scientific estimation of land demand and combined with the future development priorities of each district, the urban growth boundary will be used as a restriction area to control the amount of construction land available and reduce the other land-use types' probabilities of conversion.

Hebei: The urban growth boundary will be used as a restricted area to guide the metropolitan area of the provincial capitals to form tandem, cluster, and satellite urban spatial forms and to guide other small and medium-sized cities to create a compact and intensive layout. The probability of converting any land-use type into construction land will be reduced, and the disorderly spread of towns will be prevented.

2.3.4. Spatial Autocorrelation Analysis

The degree of aggregation or dispersion among the attributes of the spatial elements of things is reflected through spatial autocorrelation analysis, which measures whether the distribution of spatial variables is agglomerative. It is an important method and effective means by which to analyze spatial patterns. Spatial autocorrelation analysis can be divided into global and local spatial autocorrelation analysis, which are generally expressed by using Moran's I index.

1. Global Spatial Autocorrelation

$$MI = \frac{\sum_{i=1}^{n} \sum_{j=1}^{n} w_{ij} (x_i - \bar{x})(x_j - \bar{x})}{S^2 \sum_{i=1}^{n} \sum_{j=1}^{n} w_{ij}}, \qquad (4)$$

where $w$ denotes the proximity of spatial regions at $n$ locations, and $w_{ij}$ denotes the proximity of region $i$ and region $j$, $MI \in [0, 1]$. A positive value represents positive spatial autocorrelation in the attribute value distribution of the spatial thing, and the closer it is to 1, the closer the relationship between cells; a negative value indicates that the attribute value distribution of the spatial thing has negative spatial autocorrelation; 0 denotes that the spatial thing's attribute value distribution has no spatial autocorrelation.

2. Local Spatial Autocorrelation

In this study, after zoning by the district and county scales, the BTH'S local spatial clustering pattern of carbon storage was represented by the local indicators of spatial autocorrelation (LISA) [46,47].

$$MI_i = \frac{(X_i - \bar{x})}{S^2} \sum_i w_{ij}(x_j - \bar{x}), \qquad (5)$$

where $MI_i \in [-1, 1]$. Positive values indicate the spatial accumulation of similar values around the unit (high, low), negative values indicate the spatial accumulation of dissimilarity (high and low, low and high), and 0 indicates that the area has no spatial correlation with neighboring areas.

## 3. Results

### 3.1. LULC and Carbon Storage Dynamics from 1990 to 2020

3.1.1. LULC Dynamics from 1990 to 2020

The distribution and area of the BTH's land-use types over the study period are depicted in Figure 3. Cultivated land was found to be the largest land-use type, with the area accounting for more than 46%. The highest density of cultivated land, which was the dominant land-use type, was found to be in the southeast. The other land-use types, in descending order of proportion, were found to be woodland, grassland, construction land, water area, and unused land. Among them, woodland and grassland were found to be primarily distributed in the northern and western areas. Construction land was mainly found in the southeast, particularly in Beijing and Tianjin as well as along the Bohai Sea, with a concentrated area and an expansion trend.

The regional differences in the rate of land-use change can be expressed in a dynamic degree model with the following equation.

$$D = \left[ \sum_{ij}^{n} (dS_{i-j}/S_i) \right] \times 100 \times (1/t) \times 100\% \qquad (6)$$

where $S_i$ is the total area of land use type $i$ at the beginning of monitoring; $dS_{i-j}$ is the total area of land use type $i$ converted to other land use types from the beginning to the end of monitoring; $t$ is the time period; $D$ is the rate of land-use change in the study sample area corresponding to time period $t$. For convenience, it is expanded by a factor of 100.

The dynamic degree of land use in the BTH was high from 1990 to 2010, as can be seen from Table 4, and the land-use change was more drastic. After this, the dynamic degree decreased, and the land-use change tended to be moderate. From 1990 to 2020, the dynamic degree of woodland in the BTH region was 0.07%, which was the lowest value among all. On the contrary, the highest value was that of construction land, which had a dynamic degree of 3.04%. From 1990 to 2010, construction land increased greatly and urban expansion was rapid. After 2010, the expansion capacity decreased significantly. Despite the fact that the cultivated land had a relatively low dynamic degree, the area variation

remained large due to its largest area proportion. Additionally, the dynamic degree of grassland was −0.16% from 1990 to 2020. In summation, other land-use types were forced to retreat as the construction land expanded. In other words, the development of the BTH came at a certain cost to the terrestrial ecosystem. Therefore, it is of importance to assess and predict carbon storage based on LULC.

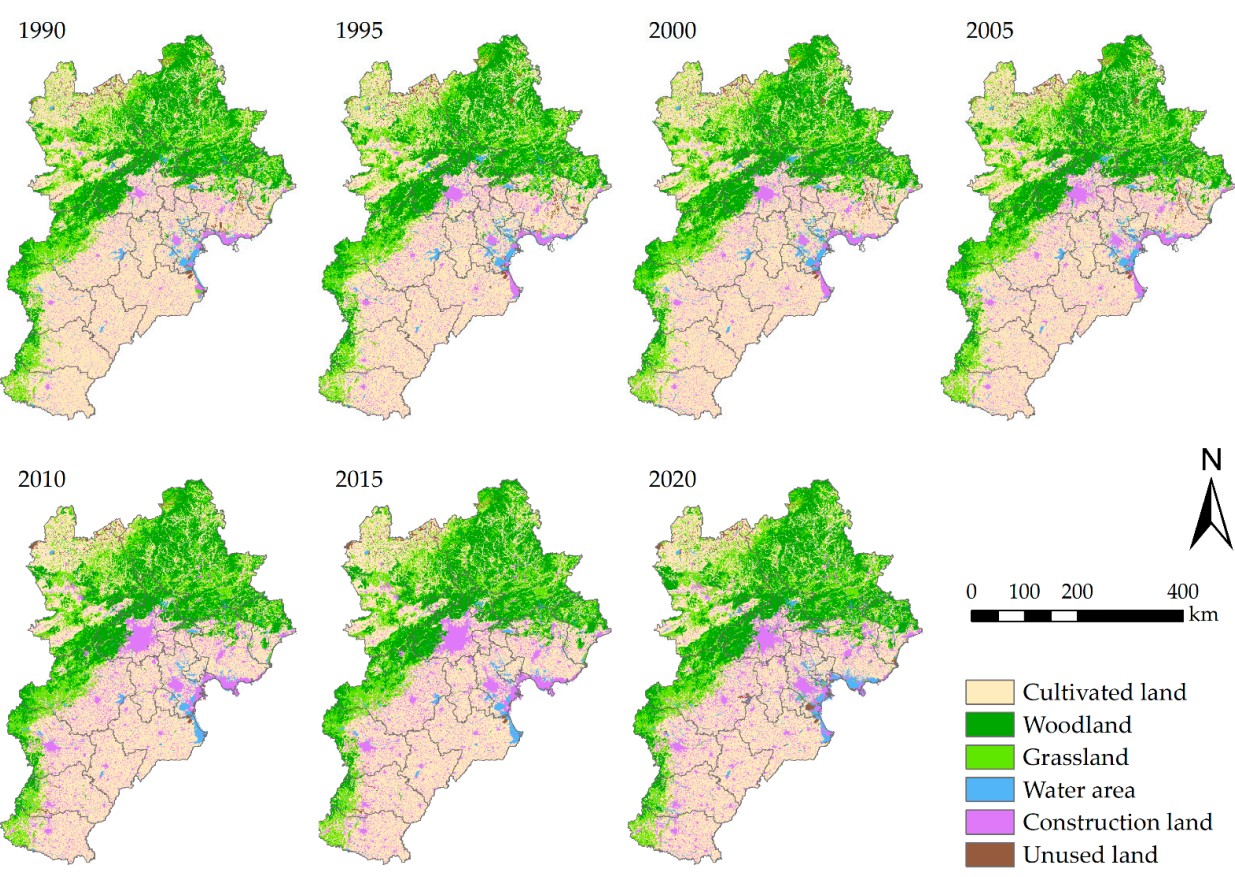

**Figure 3.** The land-use distribution from 1990 to 2020.

**Table 4.** The changes in the area and dynamic degree of land use from 1990 to 2020.

| Land-Use Type | | Cultivated Land | Woodland | Grassland | Water Area | Construction Land | Unused Land |
|---|---|---|---|---|---|---|---|
| 1990–1995 | Area Change (km²) | −2637.34 | 247.48 | −478.51 | 130.08 | 2850.58 | −162.05 |
| | Dynamic Degree (%) | −0.47 | 0.11 | −0.27 | 0.41 | 3.89 | −1.44 |
| 1995–2000 | Area Change (km²) | −262.66 | −161.47 | 69.54 | −10.58 | 343.75 | −11.06 |
| | Dynamic Degree (%) | −0.05 | −0.07 | 0.04 | −0.03 | 0.39 | −0.11 |
| 2000–2005 | Area Change (km²) | −1024.90 | 34.10 | −174.84 | −166.84 | 1517.23 | −57.63 |
| | Dynamic Degree (%) | −0.19 | 0.02 | −0.10 | −0.52 | 1.70 | −0.55 |
| 2005–2010 | Area Change (km²) | −4307.15 | 319.43 | −1077.81 | −604.02 | 6760.65 | −607.64 |
| | Dynamic Degree (%) | −0.80 | 0.14 | −0.62 | −1.92 | 6.98 | −6.01 |
| 2010–2015 | Area Change (km²) | −684.23 | −30.69 | −24.31 | −95.24 | 848.32 | −96.93 |
| | Dynamic Degree (%) | −0.13 | −0.01 | −0.01 | −0.34 | 0.65 | −1.37 |
| 2015–2020 | Area Change (km²) | −3420.88 | 549.11 | 29.23 | 1488.75 | 1069.29 | 410.56 |
| | Dynamic Degree (%) | −0.66 | 0.24 | 0.02 | 5.34 | 0.79 | 6.22 |
| 1990–2020 | Area Change (km²) | −12,337.15 | 957.96 | −1656.70 | 742.15 | 13,389.82 | −524.75 |
| | Dynamic Degree (%) | −0.37 | 0.07 | −0.16 | 0.39 | 3.04 | −0.78 |

3.1.2. Spatiotemporal Variation of Carbon Storage from 1990 to 2020

The data on carbon storage in the BTH region and their changes were estimated for seven periods from 1990 to 2020 using the carbon module of the InVEST model, as shown in Figure 4. The changes in four carbon pools were found to be generally in line with the trend of total carbon storage over 30 years, with soil carbon storage having the greatest influence on the total carbon storage value. Quantitatively, the total value in 1990, 1995, 2000, 2005, 2010, 2015, and 2020 was respectively $17.26 \times 10^8$ Mg, $17.11 \times 10^8$ Mg, $17.17 \times 10^8$ Mg, $17.13 \times 10^8$ Mg, $16.95 \times 10^8$ Mg, $16.91 \times 10^8$ Mg, and $17.03 \times 10^8$ Mg. From 1990 to 2015, there was a decreasing trend with a cumulative carbon loss of $3.5 \times 10^7$ Mg, among which the soil carbon storage decreased by $3.1 \times 10^7$ Mg. The total carbon storage increased by $1.2 \times 10^7$ Mg and the soil carbon storage increased by $0.8 \times 10^7$ Mg between 2015 and 2020. From 2000 to 2010, the regional carbon storage changed drastically and the carbon loss was serious. During this period, cities led by Beijing vigorously promoted urbanization, the regional economy was booming, and there was a higher demand for land development. After 2010, urban expansion tended to slow down, land-use changes stabilized, and carbon loss began to ease; it improved after 2015.

During the study period, the spatial pattern of carbon storage exhibited significant spatial heterogeneity in the BTH, which can be seen in Figure 5. A standard ellipse analysis was performed to discern the areas with high carbon aggregation. It's observed that the northeastern and western sections were where the areas with high carbon storage were mainly distributed, with an inverted "J"-shaped distribution. The highest carbon storage density was 190.95 Mg/ha, which was located in Chengde, northern Qinhuangdao, northwest Beijing, southeastern Zhangjiakou, and northwest Baoding; these areas are characterized by a relatively low urbanization level in the suburbs and the presence of mainly woodland and grassland. The areas with low carbon storage were mostly found in the densely populated downtown of Beijing, most of Tianjin, and the southeastern plains of Hebei Province, including Cangzhou, Langfang, Baoding, Shijiazhuang, Xingtai, and Handan, where the land-use types were mainly construction land and cultivated land, and the lowest value of carbon storage was 0 Mg/ha. Beijing and Tianjin are characterized by rapid economic development and a high degree of urbanization. The industrial structure of Hebei is singular, comprised mostly of extensive industries, and the population is concentrated. The carbon storage in these regions was found to be maintained at a low level for the substantial influence of human activities.

To better reflect the regional fluctuation of carbon storage, ArcGIS was used to calculate the differences in carbon storage between the spatial distribution maps in 1990 and 2020, and then to reclassify them. The spatial variation value of carbon storage during 1990–2020 was divided into three categories, namely decreased, unchanged, and increased, and the results are shown in Figure 6. The carbon storage variation was characterized by high aggregation and a sporadic distribution from a spatial perspective. The carbon storage was stable in most parts during 30a, as shown in the figure, and its proportion was 84.59%. Additionally, 10.31% of the area exhibited a decrease and only 5.10% exhibited an increase, both of which were sporadically distributed in the study area. Significant declines in carbon storage occurred in the central main urban areas of Beijing and Tianjin, followed by the central area of Tangshan, the western area of Zhangjiakou, the central areas of Shijiazhuang and Baoding, and the coastal area of Qinhuangdao, in which large amounts of grassland, cultivated land, and some woodland were all transformed into construction land and urban expansion was dramatic, due to which the carbon storage in their surrounding areas was radically reduced. The carbon storage of terrestrial ecosystems was found to have increased sporadically under the influence of projects such as Afforestation Construction, Grain for Green, etc. Due to the constraints of the natural geographical conditions, the levels of economic development and industrialization varied greatly among the urban areas in Hebei Province. As a result, there were large differences in carbon storage changes among regions. For example, carbon storage in cities like Chengde, Hengshui, and other underdeveloped cities was relatively stable.

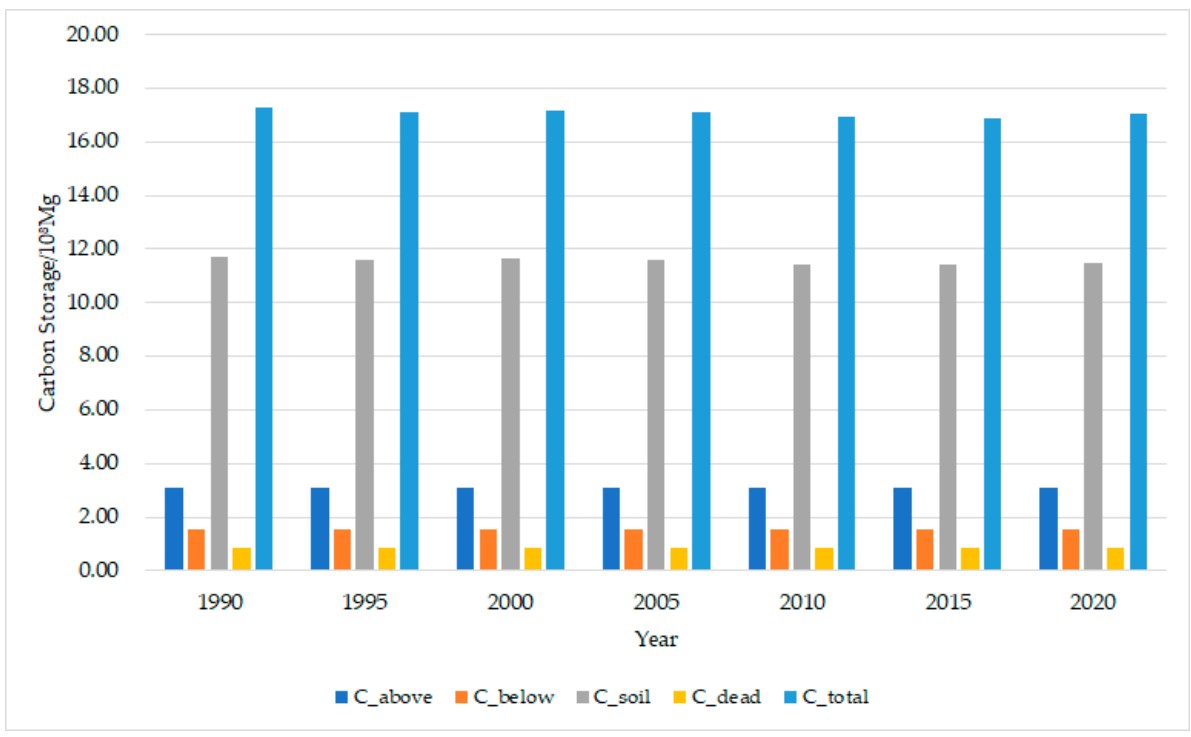

**Figure 4.** The changes in basic carbon storage from 1990 to 2020.

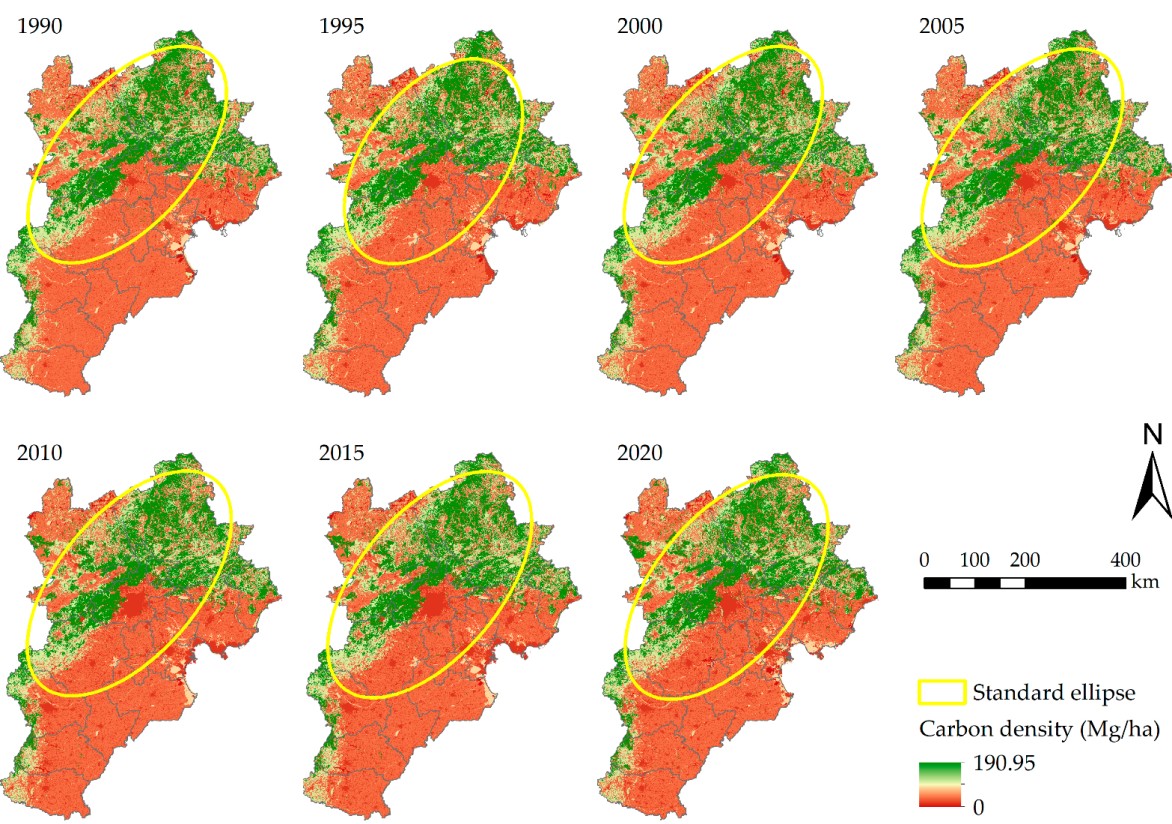

**Figure 5.** The carbon storage distribution from 1990 to 2020.

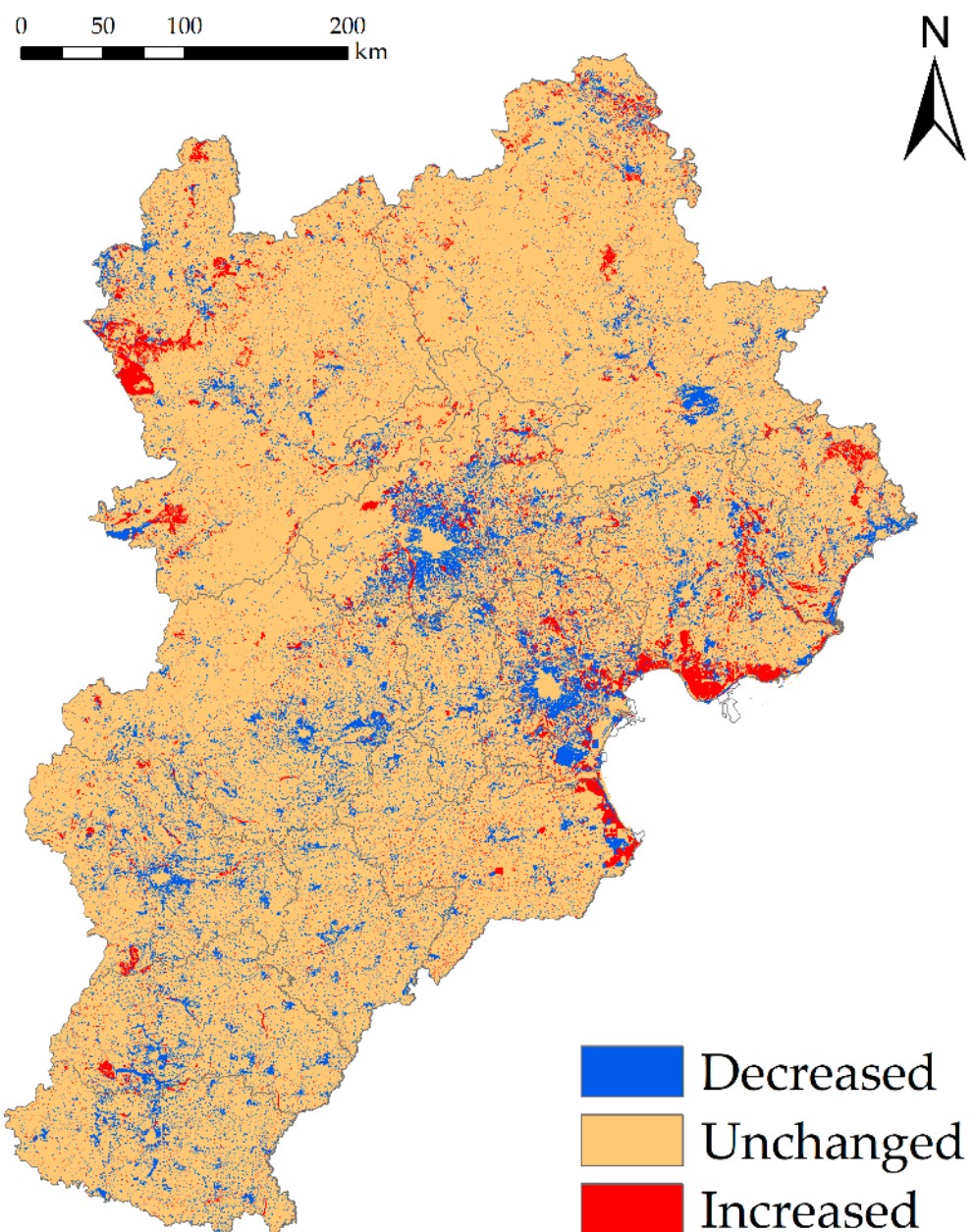

**Figure 6.** The spatial variation of carbon storage from 1990 to 2020.

*3.2. Impacts of Land-Use Changes on Carbon Storage under Different Future Scenarios in 2035 and 2050*

The data described in the previous section were processed to discern the BTH's future land use in 2035 and 2050 under four scenarios using the FLUS model, as shown in Figure 7. Under different constraints, excluding the ECS, construction land was found to continue to expand under the other three scenarios. In particular, under the EPS, the construction land was found to increase by nearly 5% in 2035 and 8% in 2050 as compared to that in 2020, with drastic urban sprawl and the destruction at the cost of ecological land. The existing cultivated land area takes up to 46% of the total area, which was found to be reduced to varying degrees under all four scenarios of future urban development. Overall, under the four scenarios, the areas with more significant land-use changes in 2035 and 2050 were found to be mainly concentrated in northern and southeastern Hebei Province and the central urban areas of Beijing and Tianjin.

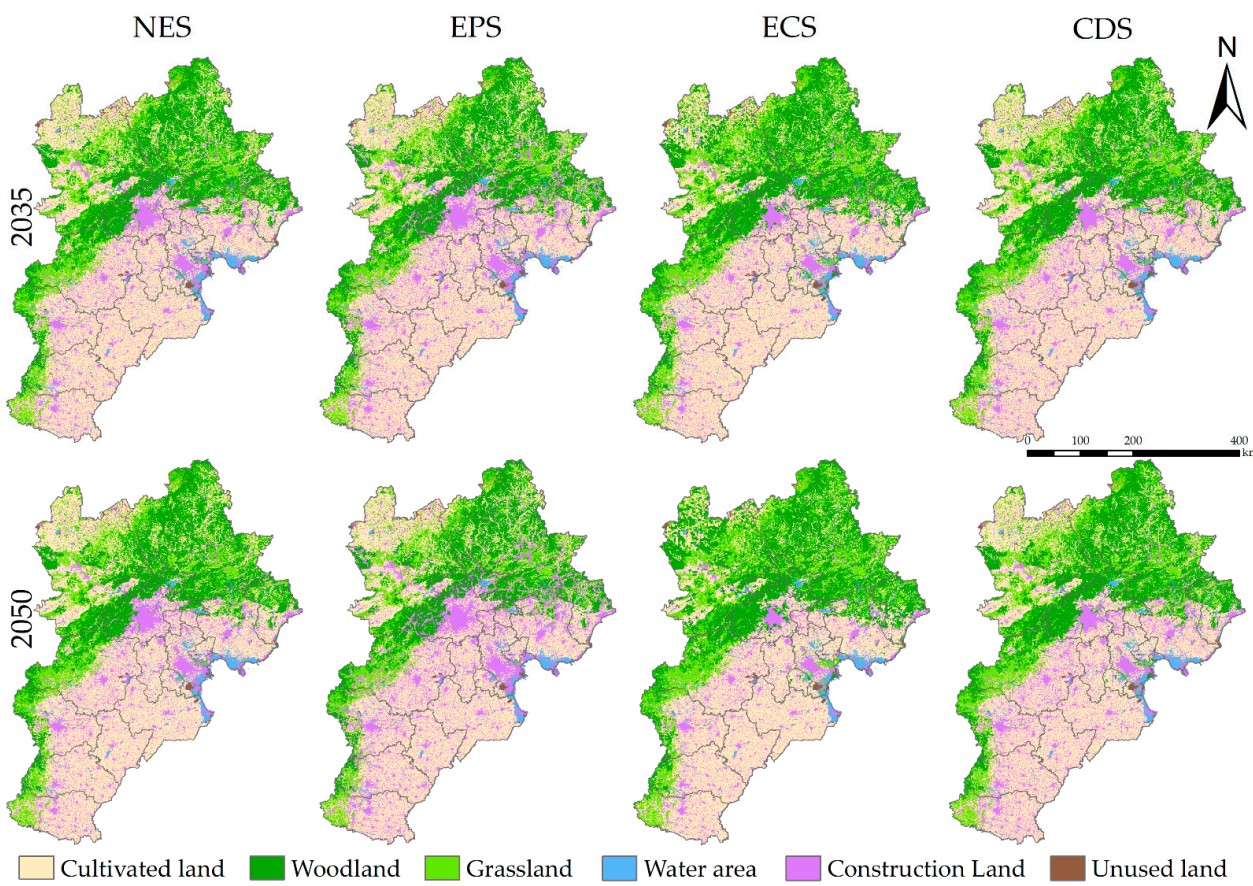

**Figure 7.** The predicted land-use distributions in 2035 and 2050 under four scenarios.

The land-use projection data were imported and run under the InVEST model's carbon module to determine the carbon storage of the BTH region in 2035 and 2050 under the four scenarios. As shown in Figures 8 and 9, the projected carbon storage data of the basic carbon pool in the BTH area in 2035 and 2050 under the four scenarios were processed. The contribution of soil carbon storage to the total carbon storage was found to be the largest, accounting for 67%, and its variation was found to be larger under the different scenarios, followed by that of the aboveground. The underground and dead organic matter carbon storage were found to account for small proportions and have relatively weak impacts on the total carbon storage. Under the ECS, the carbon storage value was found to be the highest. Conversely, among the four scenarios, carbon storage was found to reach the lowest value under the EPS.

Figures 10 and 11 present the carbon storage distributions and spatial variations in the BTH region in 2035 and 2050, which more intuitively reveal the spatial characteristics under different scenarios.

Under the NES, the carbon storage of the BTH region was found to be stable as compared with that in previous years, with the total carbon storage of $17.03 \times 10^8$ Mg in 2035 and $17.08 \times 10^8$ Mg in 2050. The northern and western woodland and grassland areas were found to have high carbon storage, i.e., the eastern Taihang Mountains, most of the Yanshan Mountains, and the areas to the north. The areas with low carbon storage were found to be concentrated in the southeastern North China Plain, which was dominated by cultivated land and construction land. Especially in Beijing and Tianjin, a small amount of carbon loss was found in the urban centers and their surrounding areas. The main reason for this may be that under natural development conditions, the ecological land in these areas has a higher probability of being converted into construction land with lower carbon density values.

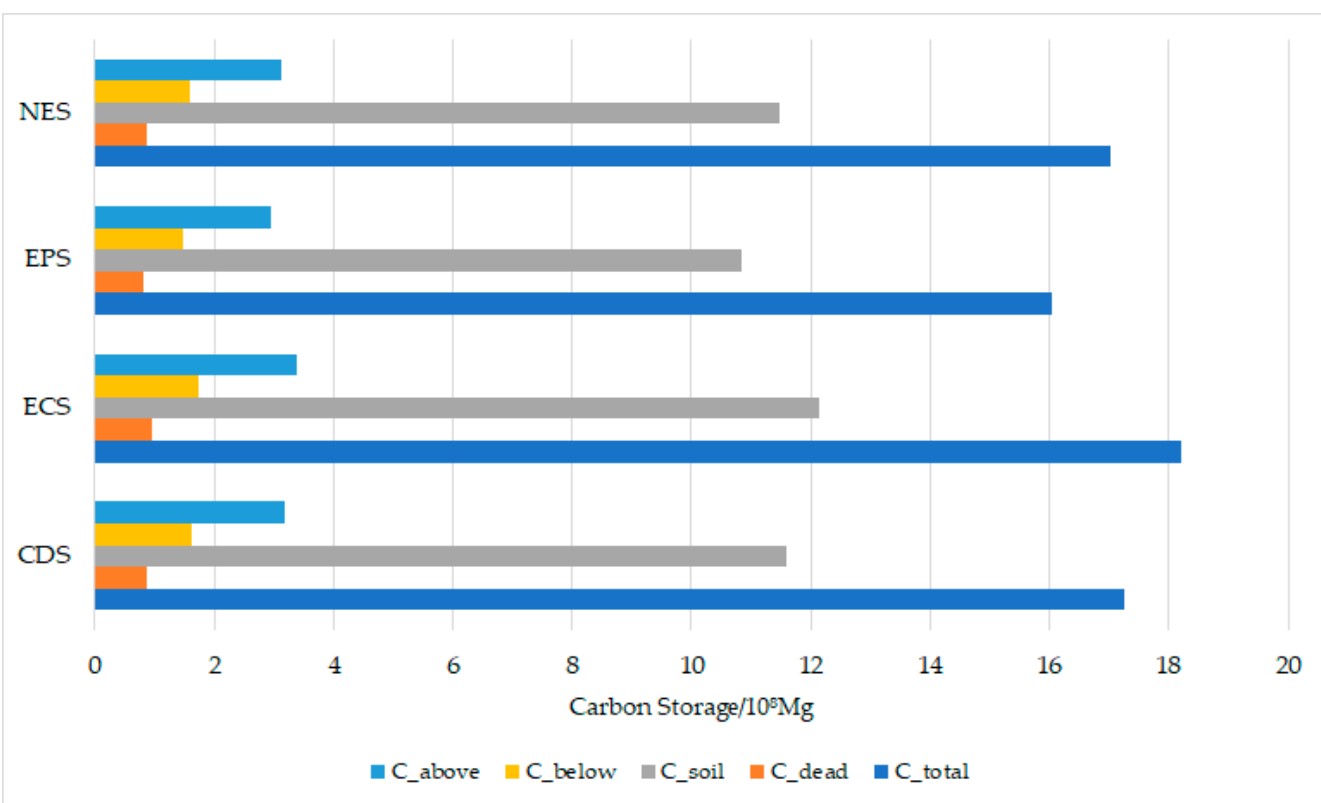

**Figure 8.** The predicted basic carbon storage in 2035 under four scenarios.

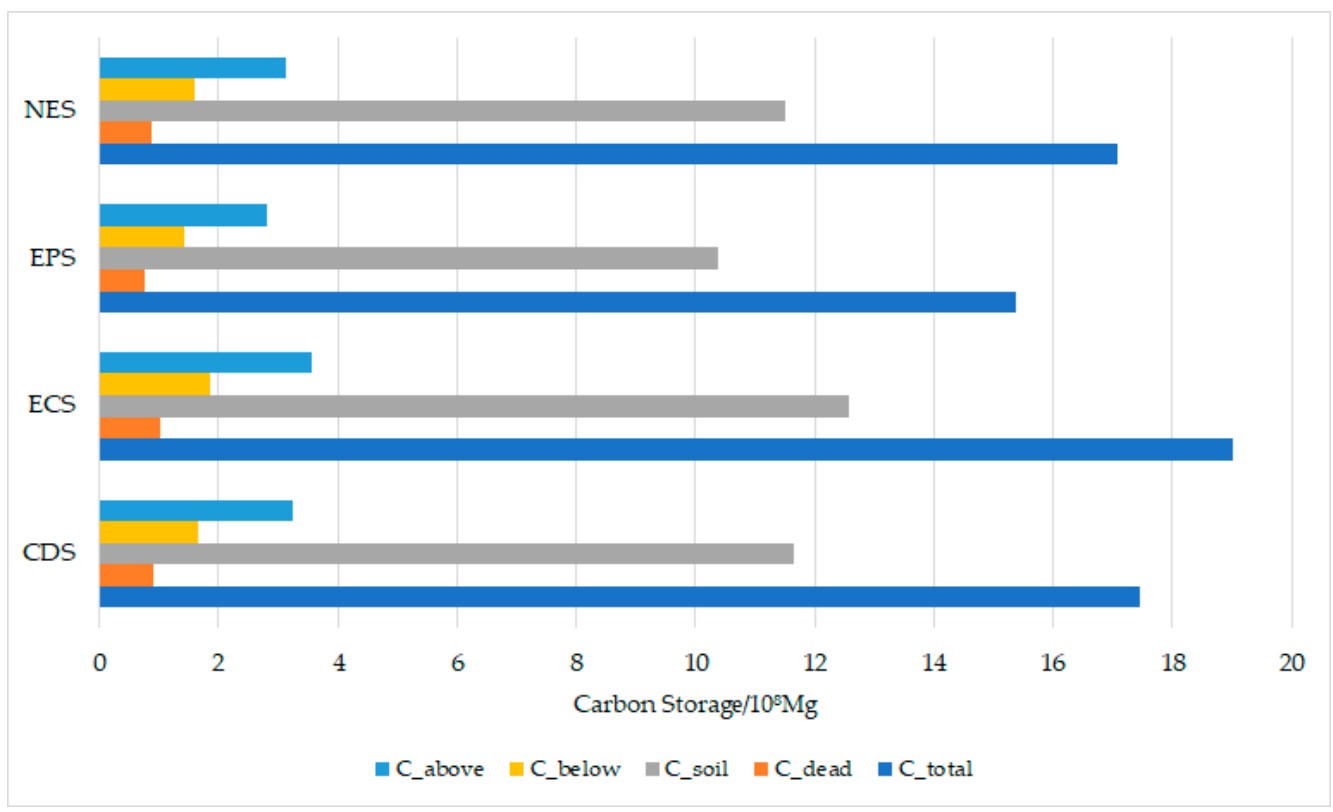

**Figure 9.** The predicted basic carbon storage in 2050 under four scenarios.

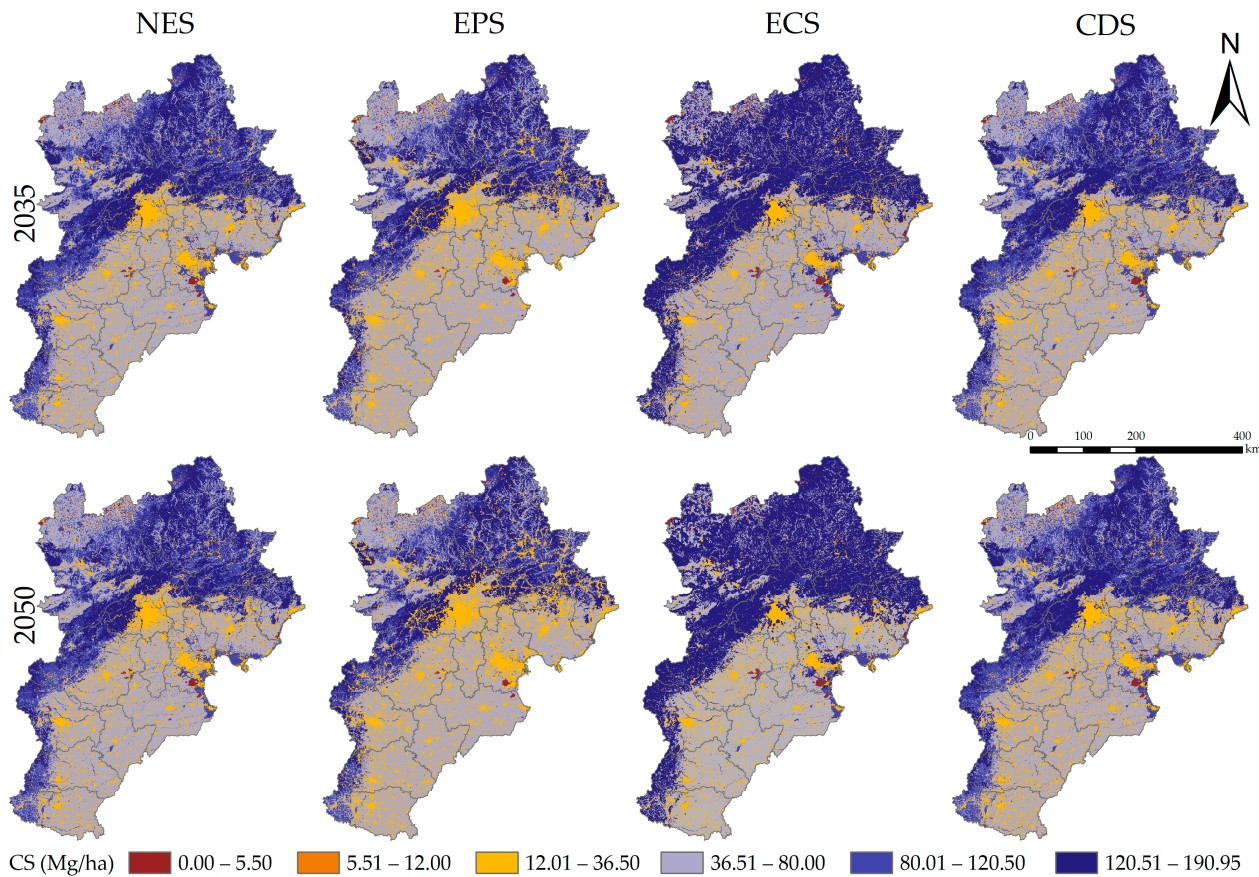

**Figure 10.** The predicted carbon storage distribution in 2035 and 2050 under four scenarios.

Under the EPS, the carbon storage of the BTH region was found to be significantly reduced and was the lowest among the four scenarios, only $15.38 \times 10^8$ Mg. With regard to land-use changes, the woodland in the north was found to be destroyed to some extent, and the spatial distribution was changed, particularly in the areas closer to Beijing, such as Chengde, Zhangjiakou, and Baoding. A large area of cultivated land in central and southern Hebei Province was found to be converted into construction land, and urban construction land in Beijing and Tianjin was further expanded. Construction land was found to be the only land-use type that increased. Figure 11 shows that sporadic decreases in carbon storage were found throughout the study area, and there were aggregated decreases in the areas around Beijing and Tianjin, as well as the northeastern section of the study area, where urbanization was found to be accelerated. In other words, the loss of carbon storage was mainly determined to be the decrease in the area of high-carbon-density woodland and grassland and the increase in the area of construction land with lower carbon density caused by the rapid growth of the urban economy.

Under the ECS, the ecological red line was strictly observed, and the expansion of urban construction land was strictly controlled. As a result, the woodland area was found to increase to 50,957.35 km² and the grassland area increased to 37,990.98 km², while the area of other land-use types decreased slightly. Carbon storage was predicted to reach the highest value among the four scenarios, rising to $18.22 \times 10^8$ Mg in 2035 and $19.00 \times 10^8$ Mg in 2050. In terms of the spatial distribution, the northern part was found to be protected and the woodland and grassland were denser, and the carbon storage increased accordingly. The urban sprawl in the central part was no longer obvious, and the conversion rate of ecological land to construction land was greatly reduced. Moreover, the construction land in Beijing and other cities was found to exhibit a trend of inward agglomeration and outward dispersion. The urban greening rate was found to be increased,

and the low-carbon storage areas were converted to higher-density areas. In contrast to the EPS, carbon storage was found to be sporadically distributed throughout the study area with increasing areas. Due to the advantage of the geographical conditions in the north, especially in the areas of Dama Mountain in the Zhangjiakou's northern part, cultivated land was found to be transformed into woodland and grassland with higher carbon density, and the carbon storage exhibited an aggregated increase.

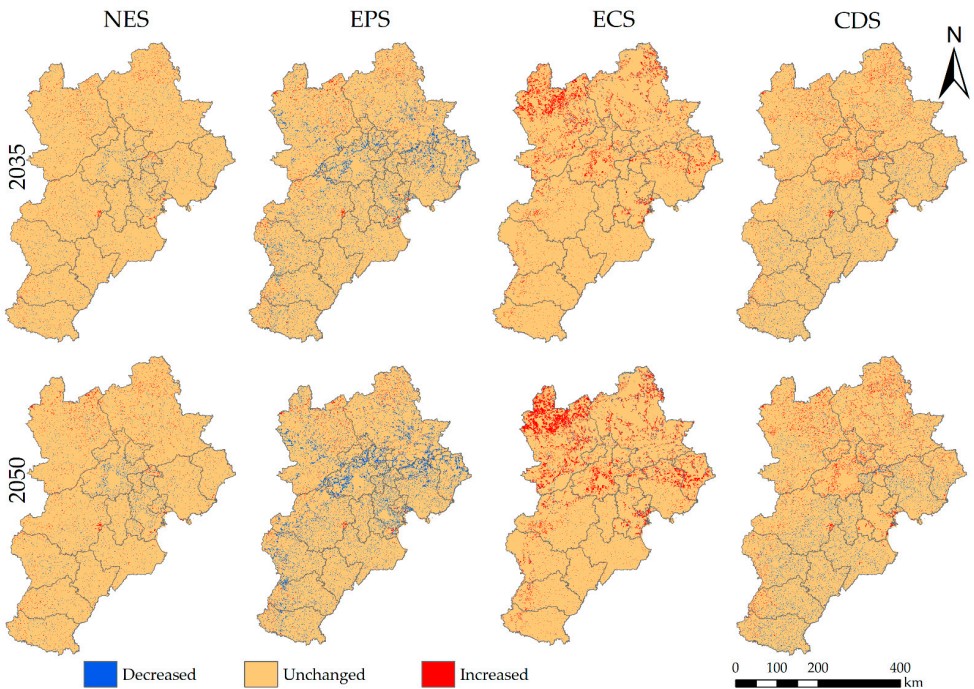

**Figure 11.** The spatial variation of carbon storage in 2035 and 2050 under four scenarios.

Under the CDS, the trend was found to be broadly similar to that under the NES. However, due to the restriction of urban growth boundaries and the reduction of the conversion rates of various ecological lands to construction lands, the carbon storage was found to increase under this scenario, with total carbon storage values of $17.26 \times 10^8$ Mg in 2035 and $17.44 \times 10^8$ Mg in 2050. In terms of spatial change, 94% of regional carbon storage was found to be basically unchanged in 2035 and 92% remained stable in 2050. It was predicted that the increase and decrease of carbon storage in 2035 and 2050 would be roughly balanced under this scenario. In addition to the increased carbon storage in the northern area grounded on the growth of woodland and grassland vegetation, a considerable increase around the periphery of the central city of Beijing was also identified. Large quantities of construction land was found to be transformed into ecological land and shifted to its periphery. In contrast, due to the decentralization of the metropolitan area, the eastern and southern areas, i.e., most of Hebei Province, were found to exhibit an increase in construction land. In particular, Langfang, Tangshan, Shijiazhuang, and other cities are actively undertaking the decentralization of Beijing's non-capital functions and making efforts to promote the Pilot Free Trade Zone's construction. Carbon storage was found to decrease correspondingly but to remain stable overall.

The carbon storage data of each city in the BTH region in 2035 and 2050 under the four scenarios were obtained by using ArcGIS to conduct zoning statistics based on municipal administrative divisions, as shown in Figures 12 and 13. The city with the highest carbon storage value in all four scenarios was found to be Chengde, which is geographically advantageous and has large amounts of woodland and grassland areas with a high carbon density. The city with the second-highest carbon storage value was found to be Zhangjiakou, which also has a low level of urban development. Beijing, the carbon storage value of which was found to be close to those of Chengde and Zhangjiakou, and

which has a dense distribution of woodland and grassland in the northwestern Yanshan and Taihang Mountain ranges, ranked third in terms of the total carbon storage value; this was due to its advantages of capital planning and policies, in addition to the EPS. Its southwestern city, Baoding, was second only to Beijing. In contrast, in Tianjin, Tangshan, Shijiazhuang, and Cangzhou, the small proportion of ecological land areas, like grassland and woodland, along with the expansion of construction land area owing to fast growing economy and urbanization, was found to lead to a relatively low carbon storage value. Although Hengshui City is economically underdeveloped and has a low degree of urban development, its overall carbon storage was found to be low partly for the dominance of cultivated land, and partly because there are few high-carbon-density land-use types like grassland and woodland.

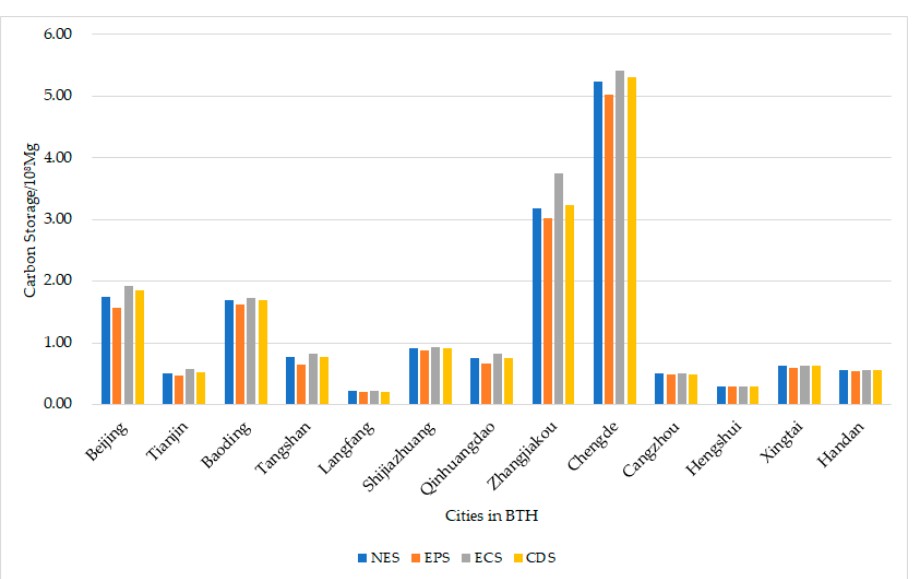

**Figure 12.** The predicted carbon storage of cities in 2035 under four scenarios.

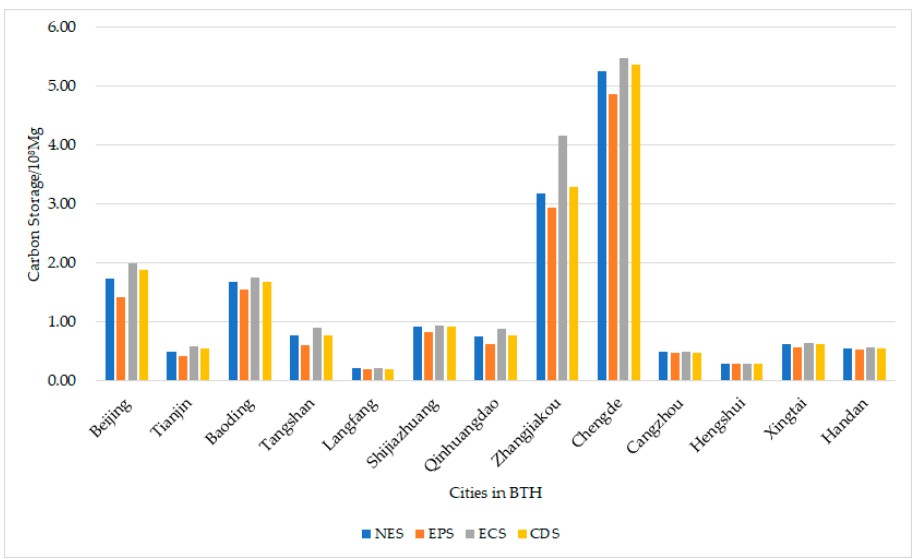

**Figure 13.** The predicted carbon storage of cities in 2050 under four scenarios.

### 3.3. Spatial Autocorrelation Analysis of Carbon Storage under Different Future Scenarios in 2035 and 2050

After processing the carbon storage in the BTH region under the four scenarios of 2035 and 2050 at the municipal scale, a global spatial autocorrelation analysis was performed

using Moran's I index of GeoDa software, as shown in Figure 14, where the blue circles indicate the cities.. The values of Moran's I were found to be 0.597, 0.605, 0.654, and 0.592 for the four scenarios in 2035 and 0.600, 0.612, 0.686, and 0.595 in 2050, respectively, which were spatially positively correlated. The strongest spatial positive correlation was under ECS and the weakest under CDS, but the general trend characteristics were found to be consistent. It indicated that the spatial distribution within cities of the BTH was not found to exhibit complete randomness, but rather spatial clustering between spatial similarities. The spatial correlation was characterized by the fact that regions with higher carbon storage tend to be adjacent to each other, and conversely, regions with lower carbon storage tend to be adjacent to each other. Most of the points were located in the first quadrant (hot spot area) belonging to high-high aggregation, and the third quadrant (cold spot area) belonging to low–low aggregation. Therefore, 12 out of 13 cities in the BTH region, except Beijing, were found to have a strong positive spatial correlation.

To reveal the local spatial clustering pattern of carbon storage, the local spatial autocorrelation analysis at the district and county scales in the BTH region in 2035 and 2050 was conducted with the local Moran's I index. The results are presented in Figure 15, which indicate that the carbon storage under the four scenarios exhibited a certain degree of similarity in spatial distribution. The high-value carbon storage areas were found to exhibit high–high clustering in Huairou District in Beijing, most counties of Chengde City, and Weixian County in Zhangjiakou City, located in the west. In the southwestern areas, Jingfu in Shijiazhuang City, Xingtai in most of Xingtai City, and Binhai New Area in Tianjin City in the east were found to exhibit high–low clustering. This is primarily due to the fact that these areas are mostly located in the Yanshan and Taihang Mountain ranges, which are characterized by high vegetation cover and the clustering and distribution of ecological land with a high carbon density, such as woodland and grassland; thus, the carbon storage values were correspondingly high. In terms of the amount of the area with high carbon storage, the largest was that under the ECS, followed by that under the CDS, and the least was that under the EPS. The areas with low carbon storage were found to be clustered in the study area's central and southern parts for the following two reasons. First, the central districts of Beijing, most of Tianjin except Jizhou District and Cangzhou City, and the eastern part of Shijiazhuang City to the south of it were found to have lower carbon storage due to their faster economic development, higher urbanization levels, and larger construction land areas. Regarding Hengshui and Xingtai, which are relatively economically underdeveloped, although their urban development is low, their carbon storage values were also found to be at a low level because their land-use types were dominated by those with low carbon density, like cultivated land and construction land.

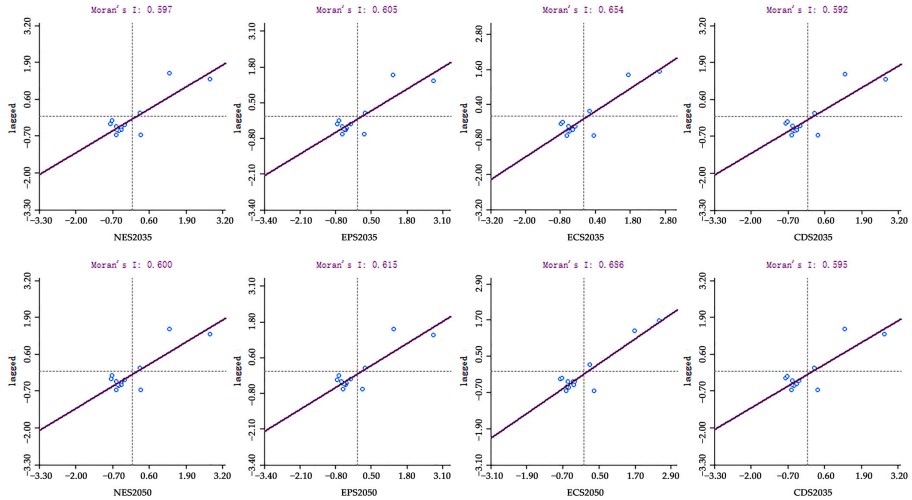

**Figure 14.** The global spatial autocorrelation analysis of carbon storage in 2035 and 2050 under four scenarios.

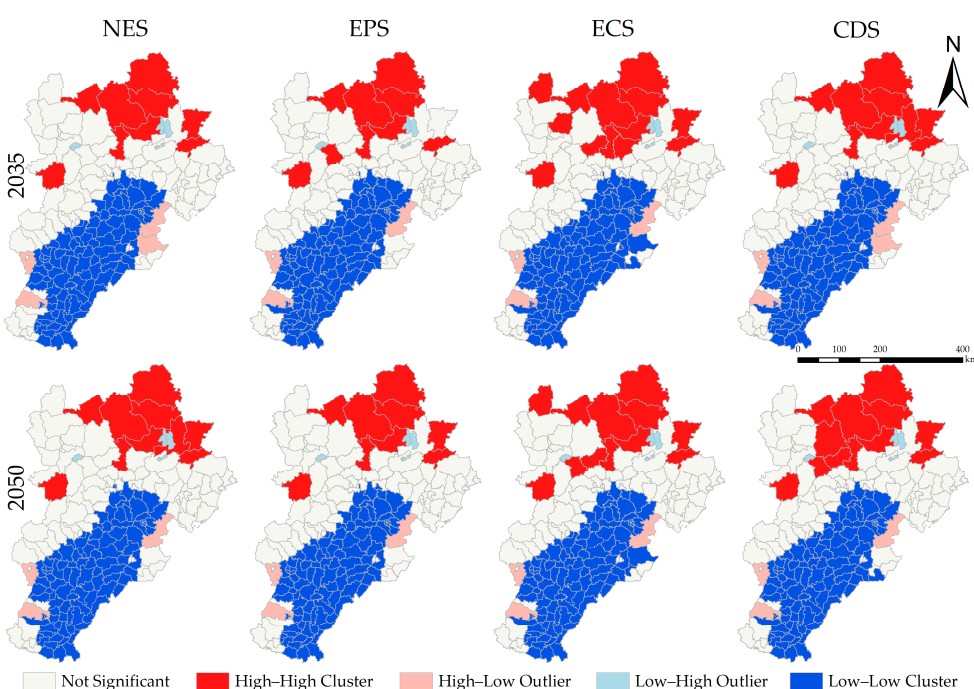

**Figure 15.** The local spatial autocorrelation analysis of carbon storage in 2035 and 2050 under four scenarios.

## 4. Discussion

### 4.1. Spatiotemporal Impact of Land-Use Changes on Carbon Storage

The carbon storage revealed an overall decreasing trend in the BTH from 1990 to 2020, which, to some extent, reflected the destruction of ecosystem service functions. In line with the results of some scholars, Wang et al. showed that carbon storage in the BTH region decreased year by year from 1990 to 2015 [29]. By simulating and projecting the distribution of land-use types and carbon storage in 2035 and 2050, this study revealed that the carbon storage was to be at risk of decreasing under the NES. The default carbon density of the InVEST model has no interannual variation, so the carbon storage changes are mainly the resulting from land-use changes, consistent with the findings of some scholars [48].

Moreover, due to the difference in the purpose of simulation among the four scenarios, the future carbon storage changes were different accordingly. The EPS ignored the sustainable development of the ecological environment and related planning policies, leading to the obvious characteristics of urban expansion. While the benefits of economic growth do not offset the negative effects of urban expansion [49]. In particular, the reduction of carbon storage was largely induced by the rapid economic development of cities such as Shijiazhuang, Tianjin and Beijing, which led to the expansion of urban construction land and the intensification of ecological land imbalance. In contrast, the EPS increased the protection of the ecological environment and strictly adhered to the ecological red lines, which was conducive to the growth and restoration of vegetation, and it can slow down or even possibly reverse the trend of environmental degradation in some regions. However, it restrained human activities to a certain extent and slowed down economic development. The CDS was not only consistent with the concept of green development but also helped to realize the coordinated development of the BTH driven by the radiation of the metropolitan area. It would be a "win–win" scenario aiming to improve the scientific foresight and decision-making rationality in the future spatial planning of the BTH.

In the coordinated development of urban agglomerations, there are extremely complex nonlinear interactions between the ecosystem as a "nutrient source" and the urban system as a "nutrient sink" [50]. Zhao et al. found that active eco-engineering measures can increase ecosystem carbon storage [25]. He et al. showed that urbanization can lead to carbon storage loss [51]. Xia et al. found that land use change from cultivated land

to industrial land was the dominant factor in the reduction of carbon sinks [52]. Zhu et al. also found that the decrease of woodland and grassland along with the increase of construction land were the major causes of the decrease in regional carbon storage in Liaoning Province, which is close to the latitude of the BTH region [38]. Land-use types, which are influenced by natural factors like topography and landscape, are the key determinants of carbon storage spatial variation. In addition, the level of urbanization is largely negatively correlated with the amount of carbon sequestration. Therefore, it is necessary to strictly protect high-quality ecological land and lower the conversion rate of construction land, so as to lead the coordinated development of land use in the BTH region by optimizing spatial patterns.

### 4.2. Expectations and Strategies

The following suggestions for the future development of the BTH region are proposed grounded on the research results.

1. Optimization of the land-use structure under the low-carbon orientation

Woodland is a core carbon pool that has a high carbon sequestration capacity and potential. In contrast, the expansion of construction land with low carbon density will pose a threat to the high-quality carbon pool. Except for the CDS, in the other three scenarios, construction land was found to exhibit different degrees of expansion.

To balance regional ecological conservation and economic growth, during the spatial planning of the country, priority should be given to the protection of dense woodland and grassland areas, mainly in the north and west parts of Hebei Province, while increasing the greening rate of other areas to maintain and improve the level of regional carbon sequestration. By combining population, economy, and industry factors, the scientific planning of various types of land should be carried out. For example, some areas in central and southern Hebei Province, which mainly include cultivated land and construction land, can be transformed from unitary production to a production-ecology complex, allowing the multiple functions of the land to be fully utilized. In addition, the increase of the construction land area should be rigorously regulated, especially in Beijing, Tianjin, and Shijiazhuang, and the carbon sequestration of construction land should be strengthened, such as by replacing impermeable surfaces in urban areas with vegetated surfaces. Moreover, a certain flexible boundary should be left according to the urban development trend.

2. Zoning management of urban patterns under the goal of coordinated development

Despite the fact that Beijing, Tianjin, and Hebei border each other, it is difficult to develop regional integration and synergy for their affiliation with different administrative units. To some extent, the capital is not confined to a single city any longer, and its functions are spread over a larger regional scale. As a result, it is necessary to rethink the urban spatial pattern of Beijing, Tianjin, and Hebei. Under the four-scenario simulations, the ECS was found to exhibit difficulty in balancing economic development and is not consistent with the direction of urban development. On the contrary, under the CDS, not only the regional carbon storage, but also the urban spatial pattern would be improved. To relieve non-capital functions, Beijing should strengthen the quality and quantity of ecological land, and the construction land should be dispersed radially from the central city outward to achieve population size reduction and function transfer in the main urban area. Tianjin's urban development has also reached a high level, and its overall pattern will no longer undergo large-scale changes. While maintaining the status quo and improving the quality of the city, the functions evacuated by Beijing will be selectively taken over by Tianjin. In Hebei Province, on account of the influence of social and natural factors, the development level varies greatly from place to place. Thus, it is highly important to adapt to local conditions; here, a few representative cities are taken as examples. Shijiazhuang, the provincial capital, is known for its high level of industrialization and a large scale of construction land expansion. While actively undertaking the decentralization of non-capital functions and industrial transfer, it should control the scale of urban expansion

appropriately and give full play to its radiation-driven role for neighboring cities like Hengshui. Chengde is constrained by natural geographical conditions and has a low level of economic development with the highest value of carbon storage. Thus, it is appropriate to plan Chengde as an ecological city. Although Zhangjiakou has a high carbon value and its expansion of construction land is not obvious, its carbon storage has been reduced due to the encroachment of woodland into cultivated land and grassland. Additionally, as one of the major cities in the ecological reserve, the regional carbon storage level of Zhangjiakou should be maintained.

3.   Energy structure adjustment and ecological compensation under an environmentally friendly policy

Human urban activities consume about 70% of the world's energy and emit nearly 80% of global greenhouse gases [53]. The BTH region has a high population density, and its energy consumption is dominated by fossil energy. Thus, the BTH region is characterized by a high energy consumption intensity and serious air pollution. Therefore, it is necessary to accelerate the adjustment of the energy structure and actively search for clean and alternative energy while improving the utilization rate of resources. For example, the BTH region is rich in geothermal resources. The development and utilization of geothermal resources can replace 87% of the total coal-fired resources, which can effectively reduce carbon emissions. In addition, due to the long-term underdevelopment of some areas, the ecological environment is degraded and the difficulty of ecological construction is increased. Thus, it is possible to build cross-regional ecological space, protect important ecological corridors, strengthen the degree of landscape connectivity, and create good base conditions for the increase of regional sinks. It is also advisable to establish a scientific and standardized ecological compensation mechanism, actively promote efficient and intensive ecological production technologies, put forth an effort in the exploration and development of new ecological agriculture, and accelerate the pace of regional industrial restructuring. Additionally, it is essential to increase public participation in the process of formulating and implementing policies and regulations associated with carbon sequestration, as well as to enhance the awareness of environmental friendliness of all people.

*4.3. Uncertainties and Limitations*

Regarding the research methods and data, the FLUS model and InVEST model were coupled in this study, thus making the most of the two models in land-use prediction and the spatial allocation of carbon storage, respectively. A more systematic and scientific research method was thus provided for the far-reaching development of carbon storage in the BTH region. However, this method was characterized by some shortcomings that require improvement.

When using the FLUS model to predict future land use, there is a limited choice of drivers, and the impact of their type and number on the simulation accuracy is unclear. Regarding socioeconomic factors, only the population, GDP, and transportation were considered. However, socioeconomic factors are complex and also include the environment, industrial structure, plant distribution, etc. With the continuation of prediction, the selection of driving factors should be experimentally investigated to the greatest possible extent to build a more perfect model.

The InVEST model assumes homogeneous and constant carbon density for the same land-use type; however, it varies with time and the environment. The carbon density coefficients of the four basic carbon pools considered in this study were obtained by drawing on previous research results and numerical experience. Due to some cognitive differences in carbon density in different regions among researchers, there may be uncertainties in the estimation results. Although this will not have a large impact on the results, the continuous field monitoring of the carbon density is expected to be carried out in future studies for the improvement in the accuracy of carbon storage simulation.

## 5. Conclusions

In this study, the impacts of land-use changes on carbon storage in the BTH region from 1990 to 2020 was estimated using the InVEST model. Then, rooted in the status quo of the spatial pattern in Beijing, Tianjin, and Hebei and relevant planning policies, four scenarios of land use and carbon storage in 2035 and 2050 were simulated and predicted by coupling the FLUS and InVEST models with the assistance of GIS. Finally, GeoDa was used to conduct the spatial autocorrelation analysis of carbon storage for future zoning management. The main conclusions are as follows.

1.  Land-use change characteristics in the BTH from 1990 to 2020 revealed that construction land exhibited the highest dynamic degree with 3.04% and cultivated land was the dominant land-use type with the largest area reduction of 12,337.15 km$^2$. The BTH showed an overall decreasing trend of carbon storage over the 30a period, with a cumulative loss of $3.5 \times 10^7$ Mg. In particular, from 1990 to 2010, the urbanization process influenced the rapid expansion of construction land and serious carbon loss. Spatially, nearly 85% of the regional carbon storage was basically stable during the 30a period, with sporadic changes in the increase and decrease.
2.  Under different constraints, the expansion of construction land was found to continue under the other three scenarios, excluding the ECS. In particular, under the EPS, the degree of urban expansion at the expense of ecological land destruction was found to be severe and to exhibit the lowest value, which decreased to $16.05 \times 10^8$ Mg in 2035 and only $15.38 \times 10^8$ Mg in 2050. Conversely, under ECS, the projected carbon storage in 2035 and 2050 reached the highest value, $18.22 \times 10^8$ Mg and $19.00 \times 10^8$ Mg, respectively. The CDS scenario was found to exhibit a trend similar to that of the NES scenario, while the carbon storage increased and the effect of relieving non-capital functions was obvious.
3.  Under the four scenarios, the spatial distributions of future carbon storage were somewhat similar, which had spatial variability. The areas with high carbon storage were found to be clustered in the northern and western parts of Beijing and Hebei Province, with an inverted "J"-shape distribution. Areas with low carbon storage were found to be clustered in the south-central part of Beijing and Hebei, as well as in most of Tianjin. With regard to the number of areas with high carbon storage, the ECS was found to be the most abundant, followed by the CDS, and the EPS was found to be the least abundant.

In conclusion, the decrease of ecological land and the increase of construction land caused by urban expansion are the prime reasons for the loss of regional carbon storage. According to the results of multi-scenario simulations, the demarcation of "three lines", including permanent basic farmland, the ecological red line, and urban growth boundary, can improve carbon storage in general. Additionally, the ecological environment should be shared while contributing to the coordinated development of the BTH region.

**Author Contributions:** Conceptualization, C.X., Y.H. and J.Z.; methodology, C.X. and Z.S.; software, Y.H.; validation, Y.H., C.X. and Z.S.; formal analysis, Y.H.; investigation, Y.H.; resources, Y.H., C.X., Z.S. and J.Z.; data curation, Y.H.; writing—original draft preparation, Y.H.; writing—review and editing, C.X. and Z.S.; visualization, Y.H.; supervision, J.Z.; project administration, J.Z.; funding acquisition, C.X. and J.Z. All authors have read and agreed to the published version of the manuscript.

**Funding:** This research was funded by the National Natural Science Foundation of China (Grant No. 72004014) and the Fundamental Humanities and Social Sciences Research Funds for the Ministry of Education of the People's Republic of China (No. 18YJC760146).

**Institutional Review Board Statement:** Not applicable.

**Informed Consent Statement:** Not applicable.

**Data Availability Statement:** Not applicable.

**Acknowledgments:** The authors are grateful to the editor and reviewers for their valuable comments and suggestions.

**Conflicts of Interest:** The authors declare no conflict of interest.

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
