# Peer review of "The Spatiotemporal Evolution and Prediction of Carbon Storage: A Case Study of Urban Agglomeration in China’s Beijing-Tianjin-Hebei Region"

_land, doi:10.3390/land11060858_

Round 1

Reviewer 1 Report

The paper presents the study of carbon storage in the Beijing-Tianjin-Hebei area. Land use cover and change simulation model and carbon storage model were created by coupling the VEST and FLUS models. The study background is well explored, the methodology is clear, and the conclusion is supported by the results. It is suggested to publish after some minor language corrections. For instance, line 48 states "in areas characterized by rapid expansion", it needs to be clarified if this is urban land use expansion or some other foot print expansion.

Author Response

Point: It is suggested to publish after some minor language corrections. For instance, line 48 states "in areas characterized by rapid expansion", it needs to be clarified if this is urban land use expansion or some other foot print expansion.

Response: The original ‘in areas characterized by rapid expansion’ in line 48 has been rectified as ‘in areas characterized by rapid expansion of urban construction land’.

Reviewer 2 Report

This appears to be a rare case where this reviewer has very few remarks.

 Are the units in Table 4 correct? No change in any land use pattern 4% in 20 years?

 Are the carbon storages given in elementary C or CO2?

Author Response

Point 1: Are the units in Table 4 correct? No change in any land use pattern 4% in 20 years?

 Response 1: It has been confirmed that the units in Table 4 are correct. Here, the land use dynamic degree is used to reflect the magnitude and rate of land change in the study area, but not the rate of area change. Related equation and instructions have been added before table 4.

The regional differences in the rate of land use change can be expressed in a dynamic degree model with the following equation.

(6)

where  is the total area of land use type i at the beginning of monitoring;  is the total area of land use type i converted to other land use types from the beginning to the end of monitoring; t is the time period;  is the rate of land use change in the study sample area corresponding to time period t. For convenience, it is expanded by a factor of 100.

Point 2: Are the carbon storages given in elementary C or CO2?

Response 2: The carbon storages are given in elementary C.

Reviewer 3 Report

There are certain sections of the manuscript that seem very easy for readers to get lost (for instance, the Introduction), and there may be opportunities to condense and streamline the writing. For instance, Line 75-108, there are too many statements on the future land-use change modeling or the carbon storage.  And the authors state “few studies have linked land use and carbon storage and conducted multi-scenario simulations of the future urban pattern to investigate the relationship”. Then, please provide the current work and focus on the summary of research gaps, objectives, or research questions.

Considering the paper is a bit lengthy. I recommend the authors move some materials from the conclusion to the Discussion and cut down the Conclusions: precise, concise, and quantitative statements about the significance of the study, highlight any new findings, and explain how the work could be extended in the future. Besides, the academic contribution is unclear.

Line 16 and Line 80: Please provide a full name for FLUS.

Please provide full names for DEM (Digital 155 Elevation Model), MAP (Mean Annual Precipitation), AMT (Annual Mean Temperature), 156, and NDVI (Normalized Difference Vegetation Index) in Table 1, too. 

Author Response

Point 1: There are certain sections of the manuscript that seem very easy for readers to get lost (for instance, the Introduction), and there may be opportunities to condense and streamline the writing. For instance, Line 75-108, there are too many statements on the future land-use change modeling or the carbon storage. And the authors state “few studies have linked land use and carbon storage and conducted multi-scenario simulations of the future urban pattern to investigate the relationship”. Then, please provide the current work and focus on the summary of research gaps, objectives, or research questions.

Response 1: The statements on future land-use simulation models in Line 75-83 have been condensed as “In future land-use simulations under different scenarios, many models have been extensively used, such as the CA-Markov model [12], the CLUE-S spatiotemporal model [13], the Future Land Use Simulation (FLUS) model [14], the Patch-generating Land Use Simulation (PLUS) model [15].” The original statement of the current work from Line 103 has been rectified as “Above all, we could conclude from the present study that land-use type change is the main driving factor of carbon storage, but current studies mainly simulated regional carbon storage by focusing on a single method or a single scenario, while few studies have explored the relationship between land-use change and carbon storage change through multi-scenario simulation of future urban pattern.”

Point 2: Considering the paper is a bit lengthy. I recommend the authors move some materials from the conclusion to the Discussion and cut down the Conclusions: precise, concise, and quantitative statements about the significance of the study, highlight any new findings, and explain how the work could be extended in the future. Besides, the academic contribution is unclear.

Response 2: Some materials has been moved from the Conclusion to the Discussion, and the Conclusions has been cut down to be more streamlined.

Conclusions:

  1. Land-use change characteristics in the BTH from 1990 to 2020 revealed that construction land exhibited the highest dynamic degree with 3.04% and cultivated land was the dominant land-use type with the largest area reduction of 12337.15 km2. The BTH showed an overall decreasing trend of carbon storage over the 30a period, with a cumulative loss of 3.5×107Mg. In particular, from 1990 to 2010, the urbanization process influenced the rapid expansion of construction land and serious carbon loss. Spatially, nearly 85% of the regional carbon storage was basically stable during the 30a period, with sporadic changes in the increase and decrease.
  2. Under different constraints, the expansion of construction land was found to continue under the other three scenarios, excluding the ECS. In particular, under the EPS, the degree of urban expansion at the expense of ecological land destruction was found to be severe, and to exhibit the lowest value, which decreased to 16.05×108Mg in 2035 and only 15.38×108Mg in 2050. Conversely, under ECS, the projected carbon storage in 2035 and 2050 reached the highest value, 18.22×108Mg and 19.00×108 Mg, respectively. The CDS scenario was found to exhibit a trend similar to that of the NES scenario, while the carbon storage increased and the effect of relieving non-capital functions was obvious.
  3. Under the four scenarios, the spatial distributions of future carbon storage were somewhat similar, which had spatial variability. The areas with high carbon storage were found to be clustered in the northern and western parts of Beijing and Hebei Province, with an inverted "J"-shape distribution. Areas with low carbon storage were found to be clustered in the south-central part of Beijing and Hebei, as well as in most of Tianjin. With regard to the number of areas with high carbon storage, the ECS was found to be the most abundant, followed by the CDS, and the EPS was found to be the least abundant.

Point 3: Line 16 and Line 80: Please provide a full name for FLUS.

Response 3: The full name of the FLUS, Future Land Use Simulation, has been added to Line 16 and Line 80.

Point 4: Please provide full names for DEM (Digital 155 Elevation Model), MAP (Mean Annual Precipitation), AMT (Annual Mean Temperature), 156, and NDVI (Normalized Difference Vegetation Index) in Table 1, too.

Response 4: The abbreviations in Table 1 have been given in table footer to the full names.

Table footer: NDVI: Normalized Difference Vegetation Index; DEM: Digital Elevation Model; MAP: Mean Annual Precipitation; AMT: Annual Mean Temperature; GDP: Gross Domestic Product; POP: Population

This manuscript is a resubmission of an earlier submission. The following is a list of the peer review reports and author responses from that submission.